# Pursuing Firm Economic Sustainability through Debt Restructuring Agreements in Italy: An Empirical Analysis

**Alessandro Danovi, Francesca Magno \* and Giovanna Dossena**

Department of Management, Economics and Quantitative Methods, University of Bergamo, 24127 Bergamo, Italy; alessandro.danovi@unibg.it (A.D.); giovanna.dossena@unibg.it (G.D.)

\* Correspondence: francesca.magno@unibg.it; Tel.: +39-035 -205-2849

**Abstract:** Corporate restructuring has become a central topic for both academics and practitioners, particularly following the global financial crisis. In particular, there is increasing interest in understanding the effectiveness of turnaround strategies, which are defined as attempts to restore the performance of firms after periods of downfall. However, despite the relevance of this issue, there is a shortage of empirical evidence regarding the effectiveness of turnaround strategies related specifically to financial interventions. Through the support of an empirical analysis among Italian firms, this paper seeks to fill this significant gap in the available literature. In particular, we conducted an in-depth analysis of 262 debt restructuring agreement (DRA) plans that occurred between 2005 and 2013 in 16 bankruptcy courts. Our study confirms the positive effect of changes in the top management team. This measure can be both a symbolic signal of genuine willingness to modify the strategy of the firm, and a real manifestation of the necessity to have new skills to complete the turnaround. In addition, the adoption of operational and strategic/asset measures increase the likelihood of turnaround success.

**Keywords:** turnaround strategy; debt restructuring agreement; global economic crisis

---

## 1. Introduction

The recent economic crisis has emphasised the need for the firms to react to unexpected events and downturns [1] in order to guarantee economic sustainability which has become an everyday worry [2]; thus, making corporate restructuring a central topic for both academics and practitioners. In particular, there is increasing interest in understanding the effectiveness of turnaround strategies, which are defined as attempts to restore the performance of firms after periods of downfall [3,4]. The importance of this topic has been further emphasised by the recent global economic crisis that severely affected many firms worldwide.

Restructuring is a complex phenomenon that encompasses many dimensions and transactions, such as selling a business division or changing the structure of capital [5]. The available literature distinguishes among four types of restructuring strategies or modes depending on the measure adopted: managerial, operational, asset/strategic and financial. In the first case, the interventions involve changes in the top management team. The second case includes measures necessary to restore efficiency and profitability (such as rationalisation of the product portfolio and reduction of costs). In the third case, the interventions are related to the assets owned by the firm or lines of business in which the firm operates (such as the sale of a business division). The final group of measures involves the financial components related in particular to the introduction of substantial modifications to the capital structure [5,6]. Examining the success of such interventions requires multi-period analyses [6].

The overall available literature is largely inconclusive in establishing the effectiveness of the four above-mentioned restructuring modes [7].

Importantly, despite the theoretical and practical relevance of this topic, there is a lack of empirical studies on this issue. There is limited knowledge based on large-scale analyses of the factors that contribute to the effectiveness of turnaround strategies. In particular, there is a shortage of empirical evidence on the effectiveness of turnaround strategies related to financial interventions [6]. Thus, through the support of an empirical analysis of Italian firms, this paper seeks to fill this significant gap in the available literature. In particular, this research focuses on the effectiveness of debt restructuring agreements (DRAs) by analysing the effectiveness of the restructuring measures (managerial, operational, asset/strategic and financial) included in such agreements. Overall, based on the analysis of 262 DRAs, this study seeks to understand whether such restructuring agreements may represent a definitive measure to guarantee firms' survival.

The remainder of this paper is organised as follows. The following section reviews the relevant literature on corporate restructuring, DRAs and their effects. Following this, we introduce the research context, explain the methods and present the results. Finally, the paper closes with a discussion of the study's implications and conclusions.

## 2. Literature Review

According to [8], the distinction between a business in financial crisis and a successfully operating business is not always evident. The former can be characterised by positive operating margins, yet excessive debt, while the latter can be characterised by negative operating margins, although the effects on the bottom line can be quite similar. [9] highlighted that, when the value of creditor claims is greater than the total liquidation value of its assets, a firm is in a distress situation. In such a context, the promptness of reaction is fundamental [1]. Nonetheless, managers often try to assume a waiting position because they dread losing their role and self-interest [6,7].

A distressed company and its creditors face two choices—restructuring or liquidation [10]—which depend on financial liquidity and credit [11,12]. According to [13], before implementing a turnaround strategy, it is important for the company to ascertain whether the going concern value is greater than the liquidation value. Otherwise, there is a serious risk of squandering resources. Turnaround strategies are then considered sets of actions and decisions able to render the crisis reversible [14,15].

In the case of restructuring, the firm can opt for different strategies characterised by different levels of depth. These strategies can range from superficial measures, such as cost reductions, to more drastic interventions, such as cutting a non-core business. The available literature highlights the existence of four major forms of restructuring: managerial, operational, asset/strategic and financial [5,6,13,16]. Making decisions about which type of intervention to implement requires an extensive evaluation of the firm's situation and the environment in which the firm operates [13,17].

Managerial restructuring is a way to stimulate changing a firm's performance through a new top management team. The rationale behind this measure is that it is difficult for existing managers to introduce drastic changes [18]. Appointing new managers is also a concrete signal to investors, banks and other stakeholders that action is being taken to modify the situation, even if the causes of the distress are not directly attributed to management [6]. The literature about the role of management change in determining the effectiveness of a turnaround strategy is unclear. Some studies underline a positive relationship between top management change and the success of turnaround strategies. According to this perspective, the new management can be better prepared to deal with the situation [19,20]. In contrast, some studies highlight the opposite effect. According to this perspective, the fear of being considered responsible for the failure pressures the new managers to adopt excessively conservative behaviour [21–23]. Nowadays, in countries such as the US, the decision to change the management is often hindered by the necessity of sophisticated distressed debt investors to access of information that only the actual management has.

Operational restructuring aims to restore efficiency and profitability through cost cutting, squandering, rationalising the portfolio of products and optimally reshaping the labour force [7,24]. In general, operational restructuring is one of the first measures adopted by a distressed firm because it is the easiest to implement. This strategy is particularly useful when the firm has problems at operational and organisational levels. This measure has the goal of stimulating profits and generating cash flow. As such, it is short-term oriented [6]. In this approach, the findings of previous studies are also mixed. In some cases, these interventions may worsen the crisis because they are oriented only to generate cash flow and profit in the short term, yet with the risk of further reducing the generation of revenue [25]. Other studies report a different effect—in particular, they highlight the effectiveness of cost reductions and of extensive supervision of stock and finance [26–28].

In contrast to operational measures, asset/strategic restructuring implies a deep reconfiguration of the firm, guided by a focus on the core business. This intervention includes strategic choices, such as asset sales, divestitures and liquidations, as well as entering or exiting joint ventures, licensing agreements and strategic alliances [16]. In particular, [27] found a substantial increase in the performance of firms that combine operational strategies (oriented to the short term) with strategic changes (oriented to the long term). However, these measures are difficult and risky to implement because the need to generate cash flow in the short term can be attained at the expense of jeopardising future development opportunities [28].

Financial restructuring involves the introduction of substantial modifications to the capital structure [5,16]. These measures are directed to find financial resources and enable the firm to go beyond the deficit of liquidity and relaunch into the market [29]. The literature stresses the existence of two different financial restructuring strategies: equity based and debt based [12,30]. The first strategy includes cutting dividends and other interventions related to equity (such as debt–equity swap and recapitalisation through credits in compensation by banks). The second strategy is related to restructuring the debt, which can include interest or principal reductions and maturity extensions [12,31]. In particular, during periods of high macroeconomic uncertainty, banks and creditors strive to de-leverage their balance sheets in the attempt to mitigate the risks of restructuring, such as clawbacks and injunctions. In our research, we focused on DRAs, which, despite having been introduced for financial restructuring, include a full set of managerial, operational, asset/strategic and financial restructuring measures.

## 3. The Research Setting

In 2005, the reform of the Italian Insolvency Law (IL) introduced a new financial instrument for turnaround—DRA. This is an insolvency approach aimed at preserving troubled companies, with the goal of discerning viable from irredeemable businesses. In detail, the 2005 reform of the Italian IL introduced the discipline of DRAs under Article 182-bis, IL, and plan validation under Article 67 IL. A validated plan is a negotiated solution that protects all acts, payments and guarantees on debtor's assets from criminal prosecution or clawback provisions, as long as the debtor is executing a plan aimed at restoring the company's debt exposure and rebalancing its financial condition.

More specifically, the Italian judicial framework has been enhanced through opening the system to the principles that inspired the United States' Chapter 11 of the Bankruptcy Code, which deals with corporate reorganisation, and the so-called London Approach introduced by the British Bank Association. In summary, a DRA is a legal procedure according to which a debtor in financial distress may pursue an agreement with at least 60% of his or her creditors, provided that the dissenting creditors are paid 'integrally' and the courts are presented with a copy of the agreements, together with a report by a certified expert ('*attestatore*') regarding the truthfulness of the data and feasibility of the restructuring plan. Thus, the DRA is configured as a two-phase procedure. First, an extra-judicial negotiation with creditors is required, with no predetermined contents. Second, a judicial approval (or rejection) by the court is required, with the related legal protection. In the event that either the prerequisites or required documents are missing, the filing is rejected.

## 4. Methods

To achieve our research goals, following a previous research [32], we conducted an in-depth analysis of 262 DRA plans that occurred between 2005 and 2013 in 16 bankruptcy courts in the north of Italy. These represented 12% of all the courts in Italy, but entailed filings that represented almost 40% of the entire population of Italian DRAs. In the 2005 to 2013 period, a total of 675 DRAs were drawn up in Italy.

In particular, we focused on the observable variables included in the DRA plans that could explain the success or failure of corporate restructuring. In addition, to corroborate our analysis, we collected other key accounting variables from the balance sheets of the firms involved in the DRAs for the time period ranging from $t - 5$ to $t + 2$, when available, where $t$ was the year of restructuring.

Through statistical analysis, we correlated the relevant managerial, operational, asset/strategic and financial restructuring measures included in the plan to the success of corporate restructuring. In detail, a successful restructuring was defined as a case in which the firm was still operating without undergoing further restructuring procedures in the two years following the examined DRA—a timeframe consistent with previous studies on turnaround strategies [6]. We used a series of chi-square tests to compare successful and unsuccessful DRAs based on the variables included in the plan. To assess the probability of success of a restructuring plan, we focused on the specific effects of relevant measures belonging to managerial, operational, asset/strategic and financial restructuring. For example, we considered sales of assets or participations; financing and new capitalisation in the form of new money; and debt–equity swaps, including participative financial instruments or credits in compensation, from the perspective of banks and shareholders. The considered variables are displayed in detail in the results section.

## 5. Sample Description

The DRAs included in our sample were mostly concentrated in the 2009 to 2013 period, as highlighted in Table 1. This concentration was strongly related to the macroeconomic conditions. After the global financial crisis, companies in distress sought legislative protection to survive as going concern. During this period a significant change was introduced in the Italian insolvency Law (called "Decree on Development") which has significant impact on the DRA procedure. However, even if the Decree became effective on August 2012, the effects of this measure will give its effects only in the long period as highlighted by [33].

**Table 1.** Temporal Distribution of the DRAs Included in the Sample.

|  | 2005 | 2006 | 2007 | 2008 | 2009 | 2010 | 2011 | 2012 | 2013 |
|---|---|---|---|---|---|---|---|---|---|
| **Frequency** | 1 | 2 | 2 | 31 | 28 | 47 | 31 | 69 | 51 |

Table 2 summarises the main characteristics of the firms involved with the DRAs included in our sample. Accounting data were related to the year of the restructuring (*t*). The secondary (43.89%) and tertiary (54.20%) sectors were the most represented. Specifically, real estate was the most affected industry (66 cases), followed by construction (47 cases) and trade (34 cases). Almost 50% of our sample comprised small firms. This is consistent with the characteristics of the Italian economy, of which the large majority comprises small firms. Another 22.14% of the sample were medium firms, while 28.62% were large enterprises.

As Table 3 shows, the large majority of companies included in the sample had experienced some sort of financial disequilibrium that naturally worsened in the years prior to the restructuring agreement. However, it was not possible to discern whether the crisis originated from operational choices, rather than financial choices.

**Table 2.** Characteristics of the Sample.

|  | **Frequency** | **Percentage** |
|---|---|---|
| Macro sector | | |
| Primary | 5 | 1.91 |
| Secondary | 115 | 43.89 |
| Tertiary | 142 | 54.20 |
| Number of employees | | |
| 0–50 | 129 | 49.24 |
| 51–100 | 29 | 11.07 |
| 101–200 | 19 | 7.25 |
| 201–250 | 10 | 3.82 |
| >250 | 75 | 28.62 |
| Debt cluster (million €) | | |
| 0–5 | 66 | 25.20 |
| 5–25 | 85 | 32.44 |
| 25–50 | 32 | 12.21 |
| 50–100 | 32 | 12.21 |
| >100 | 47 | 17.94 |

**Table 3.** Distribution of Debt–Equity Ratio.

| **Debt-Equity Ratio** | $t-3$ | $t-2$ | $t-1$ | $T$ | $t+1$ |
|---|---|---|---|---|---|
| Mean | 55.86 | 48.61 | 26.02 | 29.78 | 32.44 |
| Median | 6.62 | 6.47 | 5.54 | 5.23 | 3.83 |

## 6. Results

We performed several chi-square tests to examine whether the success of the corporate restructuring was related to the presence of specific restructuring measures in the DRA plans. Table 4 summarises the main results of the chi-square tests. Regarding the managerial restructuring strategy, the changes introduced in the top management team were statistically related to a higher likelihood of restructuring success. We then considered the effects of operational measures, such as rationalisation of the production (internalisation, externalisation and offshoring decisions) and interventions on product portfolio and labour force. In particular, we grouped the firms into two classes depending on whether they had adopted any of the six operational measures or none of them. The findings revealed that firms that used at least one of the operational measures had a higher likelihood of success. We also followed a similar procedure to examine the impact of asset/strategic measures. Firms were distinguished into two groups: those that had adopted at least one asset/strategic measure and those that had taken none of them. The results clearly highlight that the firms adopting at least one asset/strategic measure were more likely to succeed. When examining individual operational and asset/strategic measures, none of them was statistically related to the success of restructuring with the exception of 'refocusing on core business'.

In addition, several financial measures were strongly related to the positive outcome of the restructuring. In particular, regarding debt-based measures, the findings indicated that the deferral of the principal had a significant effect. It appears that, on average, plans that entailed a deferral of the principal were more likely not to file for other procedures. In contrast, the deferral of interest instalments was not significant.

In addition, the chi-square test showed that the success of restructuring occurred more often when banks provided new financing, compared with when they did not. The same did not occur when new finance was provided by shareholders. Another financial measure undertaken in corporate restructuring includes write-offs of debts (trade debts, tax debts and security debts) towards the company. However, the chi-square tests on these variables did not yield significant results. Focusing on equity-based measures, the amount of fresh equity injected into distressed companies by shareholders was statistically significant.

Overall, the findings indicated that success occurs more frequently when shareholders decide to inject new money as equity. In addition, the success of restructuring is more likely if the plans include recapitalisation through bank credits in compensation, even though these measures occur with low frequency.

**Table 4.** Results of the $\chi^2$ Tests.

| Restructuring Measures | | Restructuring Outcome (Frequencies) | | $\chi^2$ Test |
|---|---|---|---|---|
| | | Failure | Success | |
| *Managerial* | | | | |
| Changes in the top management team expected in the plan of DRA | Yes | 10 | 15 | 16.669 *** |
| | No | 184 | 53 | |
| *Operational* | | | | |
| At least one operational measure | Yes | 70 | 34 | 4.074 ** |
| | No | 124 | 34 | |
| Geographic expansion (export) | Yes | 7 | 3 | 0.089 |
| | No | 187 | 65 | |
| Rationalisation of the production: internalisation | Yes | 9 | 2 | 0.361 |
| | No | 185 | 66 | |
| Rationalisation of the production: externalisation | Yes | 9 | 2 | 0.361 |
| | No | 185 | 66 | |
| Rationalisation of the production: offshoring | Yes | 0 | 1 | 2.864 |
| | No | 194 | 67 | |
| Intervention on products in portfolio | Yes | 28 | 12 | 0.402 |
| | No | 166 | 56 | |
| Intervention on labour force | Yes | 64 | 19 | 0.593 |
| | No | 130 | 49 | |
| *Asset/strategic* | | | | |
| At least one asset/strategic measure | Yes | 114 | 52 | 6.801 *** |
| | No | 80 | 16 | |
| Refocusing on core business | Yes | 96 | 43 | 3.822 ** |
| | No | 98 | 25 | |
| Sale of real estate | Yes | 76 | 33 | 1.813 |
| | No | 118 | 35 | |
| Sale of stock | Yes | 24 | 8 | 0.017 |
| | No | 170 | 60 | |
| Sale of shareholding | Yes | 25 | 13 | 1.577 |
| | No | 169 | 55 | |
| Sale of business unit | Yes | 11 | 4 | 0.004 |
| | No | 183 | 64 | |
| Rent of business unit | Yes | 4 | 3 | 1.069 |
| | No | 190 | 65 | |
| *Financial Debt based* | | | | |
| Deferral of the principal | Yes | 65 | 46 | 24.037 *** |
| | No | 129 | 22 | |
| Deferral of interest instalments | Yes | 23 | 6 | 0.470 |
| | No | 171 | 62 | |
| New finance from banks | Yes | 14 | 14 | 9.432 ** |
| | No | 180 | 54 | |
| New finance from shareholders | Yes | 16 | 9 | 1.451 |
| | No | 178 | 59 | |
| Write-offs of trade debts | Yes | 27 | 14 | 1.697 |
| | No | 167 | 54 | |
| Write-offs of tax debts | Yes | 13 | 5 | 0.033 |
| | No | 181 | 63 | |
| Write-offs of social security debts | Yes | 7 | 0 | 2.521 |
| | No | 187 | 68 | |
| *Equity based* | | | | |
| Fresh equity by shareholders | Yes | 14 | 13 | 7.715 ** |
| | No | 180 | 55 | |
| Recapitalisation through credits in compensation by banks | Yes | 3 | 7 | 10.495 *** |
| | No | 191 | 61 | |

** $p < 0.05$; *** $p < 0.01$.

## 7. Discussion

The findings of our research contribute to advancing knowledge regarding the effectiveness of turnaround strategies. Until now, the effects of specific restructuring interventions related to changes in the top management team had not been clarified, with the available literature highlighting the existence of mixed effects. Some studies [21–23] reported a negative effect, especially because new managers dread being considered responsible for the failure, and subsequently do not adopt adequate strategies. In addition, changes among top managers can create further instability within the firm, especially in terms of the security of the labour force [20]. In contrast, some studies supported the existence of a positive relationship between the change of top managers and the success of restructuring interventions [19,20]. Our study confirms the positive effect of changes in the top management team. This measure can be a symbolic signal of the willingness to genuinely modify the strategy of the firm but also a real manifestation of the necessity to have new skills to complete the turnaround. In fact, the likelihood of success in restructuring is also related to the operational, asset/strategic measures and financial restructuring measures adopted by the firm.

Our findings contribute to solving inconsistencies in the previous literature regarding operational restructuring measures [6]. According to some authors, operational measures may intensify the crisis [7,25], while other studies reported the opposite effect [27,28]. Our research shows that operational interventions are related to the success of the restructuring process. At the same time, the findings highlight that there is not one individual operational measure which is statistically associated to a higher likelihood of the success of the turnaround. This result suggests that the type and mix of operational measures to be adopted should vary depending on the specific situation of the firm.

In addition, consistent with studies that based the success of turnaround on asset/strategic measures [6,18], our research found that strategic interventions are related to the positive outcome of the restructuring. However, as for operational interventions, the types and mix of asset/strategic measures for each firm should be carefully selected depending on the situation [28,34].

Finally, our study corroborated the importance of financial restructuring measures. However, consistent with the available literature [35], write-offs of debts did not exert statistically significant effects. In particular, our findings emphasise the importance of debt-based strategies related to deferral of the principal. In addition, this research highlights the role of banks in successful restructuring in terms of introducing new finance, but also whenever banks are investors in the shareholding. This is consistent with the idea that the capital structure of the company, together with the number of credit banks involved, is a key determinant of the success of a plan [36].

## 8. Implications and Conclusions

Crises can have devastating effects for firms: they can alter the firms' priorities and the relationships with their stakeholders. In the worse cases they can even threat the firms' survival. When a crisis occurs, it is first of all important to understand the origin and the timing of the downfall. Only by gaining this awareness, the company may be able design proper interventions. On this point managers should note that in many cases the crisis results from a combination of internal (increased competition, economic crisis, etc.) and external (failure in new products, wrong investments etc.) factors. Following this perspective, turnaround strategies must be conceived as means to remove the causes of the crisis and to lead the firm back to a sustainable performance

Through the support of an empirical analysis, this study contributes to advancing knowledge about the effectiveness of such turnaround strategies. As the available literature has underlined, there is a lack of knowledge based on large-scale evidence. This study contributes to closing this gap through analysis of the effects of restricting measures included in 262 DRA plans that occurred between 2005 and 2013. In addition, to corroborate our findings, we collected variables from the balance sheets of the firms involved for the time period ranging from $t - 5$ to $t + 2$.

Overall, our study suggests the importance of changing the top management team. The relationship between new managers and the higher likelihood of success of the turnaround can be explained

through the operational, asset/strategic and financial measures taken by the new managers. In fact, all these interventions have significant impacts on the success of restructuring. At the same time, none of the operational and asset/strategic measures (with the exception of 'refocusing on core business') is able alone to statistically explain the success of the turnaround. This finding suggests that new managers may be more likely (then previous managers) to identify the right mix of the operational and asset/strategic measures required by the firm.

In addition, our results contribute to the turnaround research and practice in two ways. First of all, it is evident that turnaround strategies will be effective if the necessary sacrifices are balanced among the stakeholders. As a consequence, we can interpret the results in the light of stakeholder theory [37]. According to this approach, the success of the firm depends on the supports it receives from its stakeholders. Therefore, in the case of turnarounds, it is important to understand the perceptions and the expectations of the stakeholders. From this perspective, the most important stakeholders should be involved to obtain the consensus around the measures adopted because they supply the resources needed to face the crisis situation. In this regard, it should note that creditors are inclined to accept the deferral of the principal if shareholders insert fresh equity and there is a recapitalisation through credits in compensation by banks. Second, our study underlines the importance of the leadership change. Managers are often considered the causes of firms' crisis and in many cases, they are too involved and uncritical to really understand the causes of the downturn and to adopt the correct solutions. On the contrary, changes in the top management team provide the firms with new skills, beyond giving a signal of the firm's willingness to genuinely complete a turnaround and ensuring the continuity of the enterprise [38].

Finally, while this paper examined firms that are already facing severe crises, the importance of recognizing early signals of future crises must be stressed. Managers should design a dashboard that includes a number of performance indicators and clearly identify who is charge of monitor the trends and make the related decisions [39]. Overall, it is clear that enhancing the managerial and entrepreneurial education of the top management team is a fundamental way to better prevent and deal with the crises.

Finally, it should be noted that this study presents several limitations. First, the analysis was based on data collected only from Italian firms. Moreover, even though this research considered around 40% of all Italian DRAs, the total number of DRAs was still limited. Therefore, it would be useful to extend this research to other countries and larger samples. In particular, it would be interesting to compare the effects of the restructuring measured adopted under DRAs and Chapter 11. Finally, evaluating the combined effects of some of the restructuring interventions may represent another fruitful avenue for future research.

**Author Contributions:** Conceptualization: A.D., F.M. and G.D.; Data curation: A.D.; Formal analysis: F.M.; Investigation: A.D.; Methodology: F.M.; Supervision: G.D.

**Funding:** This research received no external funding.

**Conflicts of Interest:** The authors declare no conflicts of interest.

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
