# Peer review of "Pursuing Firm Economic Sustainability through Debt Restructuring Agreements in Italy: An Empirical Analysis"

_sustainability, doi:10.3390/su10124830_

Round 1

Reviewer 1 Report

The presented work starts from a good objective but the applied methodology is very basic, which does not provide a relationship between the variables, nor the effect of several variables together.

In reference to the conclusions and due to the applied methodology they are also very basic and should be deepened more in them

- the methodology used, the statistical tools, are improvable. We must try that the causality relationship is not only taking the variables independently, but we must see their joint effect on the main hypothesis. It can be used as for example the methodology of structural equations

- we must go deeper into the conclusions and demonstrate those that are cited but that are not justified in the development of the work

Reviewer 2 Report

Dear authors, in my opinion you have a very nice paper, well structured and easy to read. This restructuring phenomenon is very complex and not homogeneous between diferent countries. You justify very well the Italian case selection.

Your literature review is not very deep but it is interesting and it presents very well the global idea.

In your sample descrition you have a minor error (lines 196-197). The primary sector is not one of the most represented.

I have some concerns about your methodology, and the use of chi-square test. For each chi-square test, if n> 20 there can be no more than 20% of the cells with expected frequencies below 5 nor can there be any cell with an expected frequency of less than 1. For each test you have 4 cells, so none can be in this situation. You may test the expected frequencies and, then, if neccessary, use other tests.

You claim that “In addition, this research highlights the role of banks in successful restructuring in terms of introducing new finance, but also whenever banks are investors in the shareholding. This is consistent with the idea that the capital structure of the company, together with the number of credit banks involved, is a key determinant of the success of a plan.” (lines 278-281). It looks to me inconsistente with your results (your last chi-square test).

I don’t understand the basis to claim that: “Finally, the results suggest other practical insights. First of all, firms should adopt and implement measurement systems that allows them to recognize weak  signals of future crises in advance. In many cases the problem with measurement systems is represented by the fact that many ratios are adopted and monitored but without clear routines and procedures to link them to the decision-making process. On the contrary, it is important to design a dashboard that includes a limited number  of indicators, to establish the target level of performance to be accepted for each of them and to clearly identify who is charge of monitor the results and make the related decisions [37]. Overall, it is clear that enhancing the managerial and entrepreneurial education of the top management team is a fundamental way to better prevent and deal with the crises.” (liines 321-331). I dont´t understand how results suggest this.

In my opinion you didn’t evaluate the distinct effects as you say in line 338.

I don’t know what are you talking about in line 339 when you refer chapter 9.

Round 2

Reviewer 1 Report

I believe that the response to the initial comments as well as the modifications presented are acceptable and I thank the authors for their effort to improve the work presented.

Reviewer 3 Report

I like the effort to diagnose the determinants of successful turnaround and find this paper provides relevant insight into this area. I am pleased with the revisions which addressed my concerns/confusions from my first reading of the paper, and find the analytics and the results much more intuitive and consistent.

Sustainability EISSN 2071-1050 Published by MDPI AG, Basel, Switzerland RSS E-Mail Table of Contents Alert
Back to Top